# An Industrial Case Study on Explaining Risk of Code Changes using LLM-based Prediction Models

## Abstract

Predictions by machine learning (ML) and artificial intelligence (AI) models are often received skeptically unless they are paired with intelligible explanations. In the context of just-in-time defect prediction, highlighting small portions of a software change (*diff*)—beyond rule-based lints—where risk may be concentrated has not yet been extensively investigated. In this work, we leverage attention weights from an LLM-based Diff Risk Score (DRS) model to highlight parts of a diff that the model focuses on when predicting risk. We aggregate token-level attention into interpretable code units (lines, hunks, and files), and present the top-$K$ units to developers as a lightweight form of guidance during code review. We evaluate our approach using expert-labeled changes that have caused real outages. Results show that the highlighted snippets cover expert-labeled outage-causing change lines 53.85% of the time when highlighting the top-2 hunks, while requiring developers to review 26.28% of the changed lines on average. Because attention is produced during standard model inference, the approach is scalable for large development workflows and can be surfaced in the code review UI with low additional latency.

## Keywords

Code Risk Score, LLMs, Explainability, Applied Research.

**ACM Reference Format:**

Anonymous Author(s). 2026. An Industrial Case Study on Explaining Risk of Code Changes using LLM-based Prediction Models. In . ACM, New York, NY, USA, 8 pages. https://doi.org/10.1145/nnnnnnn.nnnnnnn

## 1 Introduction

In the (big) tech industry, software quality has a direct impact on revenue, and various tools are utilized to help maintain quality to the highest standards. A just-in-time defect prediction system (e.g., *Diff Risk Score* (DRS) [1, 2]) is employed by some companies to predict potential issues in code changes. This type of system produces a risk score indicating the likelihood of a service outage if a particular code change is landed. If the risk is deemed high enough, developers may be prevented from landing the change, or required to take additional steps (e.g., adding tests, requesting additional review, or performing further validation).

We have historically explored the use of logistic regression (LR) models for gating risky diffs (e.g., [2, 3]). These LR models are

trained on curated and intuitive measures, such as the number of code review comments and the author's experience with that part of the codebase. Logistic regression is a simple causal model where each predictor has a coefficient that shows if, and by how much, it increases or decreases the risk. It has been argued that causal models are necessary for explainability [4, 5]. More complicated ML models can only tell how important a feature is to explain the variation in the response, but lack this simple mechanism provided by logistic regression [6, 7]. Researchers have introduced entire toolkits to support traditional feature-based explainability methods [8–10].

A significant limitation of the LR model is its inability to fully utilize the textual content of a given diff (such as the raw code changes themselves). In contrast, the emergence of large language models (LLMs) has opened possibilities to employ textual and other information for risk prediction. Recent work in the software engineering community, such as Diff Risk Score (DRS) [1], has shown the promise of using state-of-the-art LLMs for gating risky code changes. By risk-aligning LLMs through fine-tuning on code changes, summaries, and titles, DRS was able to gate risky diffs with significantly higher accuracy, capturing over 40% of outages by gating just the top 10% of diffs by risk [1]. It is important to note that diffs causing outages are incredibly rare (approximately two in a thousand) and prediction models tend to perform poorly. Hence, inherently, the coverage is low and developers need as much explanation and justification in order for them to seriously pay attention to flagged diffs.

Despite the higher accuracy of LLMs, engineers demand to know why their diffs are considered risky [11]—a risk score alone is not *actionable*. This need for an explanation is a major obstacle to the use of LLMs for risk prediction. The lack of explanation is particularly problematic for larger diffs, which might contain multiple large files. In addition, model errors are unavoidable: even a correct high-level warning can be ignored if it does not help an engineer narrow down what to inspect, test, or reconsider.

In this work, we present a practical method to make LLM-based risk scores more interpretable by leveraging aggregated attention. When the model produces a risk score, we highlight which parts of the diff it attends to most strongly. The intent is not to claim a causal explanation; rather, we provide a low-latency signal that can guide inspection by narrowing attention to a small number of code regions likely to have influenced the prediction.

Our initial results show that our method covers risky lines in the diff that actually caused faults 53.85% of the time when highlighting the top-2 hunks. By providing transparent and interpretable highlighting, our approach aims to address a major challenge of utilizing LLM-based risk models at scale, where the responsibility of determining what should and should not be released falls to the engineers writing and reviewing the code. Beyond increasing trust and adoption, attention-based explanations are highly scalable since attention is generated as a part of LLM inference. Our

key contribution is to develop and evaluate an approach to explain just-in-time defect prediction via LLMs to professional software engineers in a way that may improve both trust and usability. This study is done in a large-scale industrial context at FAANGCompany.

The remainder of this paper is organized as follows. Section 2 discusses FAANGCompany's development practices and explainability challenges. Section 3 details the use of LLMs and attention weights to highlight risky code segments. Section 4 covers the datasets, metrics, and experimental results on explanation effectiveness as well as user feedback. Section 5 covers related work on explainable AI in defect prediction. Section 6 discusses the threats to the validity of this work, and Section 7 concludes with a summary and discussion of future directions.

## 2 Background

### 2.1 Developing Software at FAANGCompany

At FAANGCompany, software engineers submit tens of thousands of pull requests (or "diffs") every day. Each diff goes through extensive automated testing, code review, canary deployment, and more to maintain a high level of code quality and reliability in our systems[1]. While many diffs are routine, a small fraction can introduce severe incidents due to subtle interactions, incomplete validation, or misconfigurations. When systems do break, we track and report these incidents as outage (or "outages") as part of our incident response process. In this work, we focus on outages that can be attributed to a particular landed diff, enabling root-cause analysis at the level of changed lines.

Diffs can—and do—cause outages. During major revenue events, like the holiday season, we used to conduct "code freezes" in which we disallowed all diffs from landing in an effort to minimize outages. This lowered risk at the cost of developer productivity. Recently, we have moved away from this practice toward a more dynamic risk management system that allows us to balance productivity and risk [1].

### 2.2 Diff Risk Score (DRS)

Diff Risk Score models generate a score between 0 (no risk) and 1 (maximum risk) for diffs before they land [1, 2]. The score can then be used to allow the lowest risk diffs to land or block the highest risk diffs from landing (with mechanisms for exceptions in emergencies), depending on the desired productivity-risk trade-off. In practice, teams may tune thresholds and policies (e.g., block above a certain score, require additional review, or require additional testing) based on their tolerance for risk and operational constraints. Figure 1 shows what developers see in the code review UI. It includes the risk score of the diff, reasons why diff is considered risky, and potential next actions. It also includes a feedback mechanism to let engineers provide feedback for future improvements. Our goal is to strengthen this UI so that engineers can quickly determine *what* to inspect when a diff is flagged.

---

[1]What is it like to write code at FAANGCompany? — FAANGCompany Tech Podcast, Episode 55, https://engineering.fb.com/2023/09/05/web/what-like-ship-code-meta-tech-podcast/

### 2.3 Explaining Risk

The DRS toolset is designed to be actionable and transparent. It provides feedback on why a diff received a particular risk score and suggests actions engineers can take to reduce that risk, like improving test coverage or refactoring code, as shown in Figure 1.

A score alone is insufficient to achieve this objective. As an AI/ML based approach, DRS is often seen as a black box by software engineers. For many cases, users are informed that a diff is dangerous with little justification or context. This is especially the case when using LLM-based DRS classification as opposed to logistic regression based classification, where the latter offer some explainability through their coefficients and feature values.

The lack of good explanations presents major barriers to the adoption of DRS:

- **Reduced User Productivity:** Without clear instructions, engineers struggle to unblock themselves from landing high-risk diffs, compromising developer productivity.
- **Reduced Trust:** Inaccurate results without rationale drive skepticism and decreased attention to warnings.
- **Harder Debugging and Maintenance:** Debugging DRS without rationale is like tuning a black box, making maintenance and improvement difficult.

## 3 Approach

To tackle the above issues, we highlight parts of the diff for engineers to inspect when a diff is flagged as high risk. In contrast to feature-based explanations that summarize *why* a diff is risky in terms of metadata, we focus on *where* the model attends within the diff content itself.

### 3.1 Overview

Our approach, depicted in Figure 2, uses an LLM to evaluate the risk of a code change (diff) potentially causing severe issues. The diff is embedded into a pre-defined prompt, tokenized, and fed into the LLM, which produces a risk score based on a single logit output. During this inference, the transformer computes attention weights between the generated output token and the input tokens. We aggregate these attention weights to produce a token-level importance vector, then map token importances back to code units (lines/hunks/files). We present the top-$K$ units to developers as a compact "review budget" intended to reduce time spent scanning large diffs. Because attention is computed during standard inference, highlighting adds minimal overhead and is feasible to deploy at scale in high-throughput code review workflows.

### 3.2 LLM-Based Diff Risk Score Model

To generate risk scores using LLMs, we apply the prompt demonstrated in Figure 1. As shown in Figure 2, we instruct the LLM to generate a one-token response (0 or 1) for whether or not the diff could cause an outage. We then take the tokens' logits as the risk likelihood. Concretely, the model produces scores for both output tokens; we transform these logits into a single risk score that can be thresholded by downstream policy (e.g., block, require additional review, or warn). The model used in this work leverages a LLM instance [12] which is instruction fine-tuned on historical diffs, similar to [1]. This design allows deployment with a flexible

**Figure 1: A redacted Code Review Tool UI showing risk and explainability of an outage (referred to as "SEV" in the image).**

gating threshold to adapt to different base rates of outages across products and infrastructure components. It also enables continuous calibration as the risk distribution changes over time. The LLM model we leverage in this work is the Llama3-70B [1].

## 3.3 Explanations With Attention Weight

Leveraging transformer attention weights for the purpose of explanation has been widely discussed [13–16]. In this work, we use attention as a practical proxy signal to highlight portions of the input the model attended to when producing its classification output. Our objective is to provide actionable guidance (what to inspect) while avoiding free-form natural language rationales that may hallucinate.

*Token-level attention extraction.* Given a prompt with diff content of length $N$, the LLM generates a response of length 1. We obtain the last-layer attention tensor for the generated token with shape $\{H, 1, N + 1\}$ where $H$ refers to the number of attention heads and $N + 1$ is the length of input prompt plus the generated one token response. We then pool across heads by averaging to obtain a vector $w \in \mathbb{R}^{N+1}$, where $w_i$ denotes the attention weight assigned to the $i$-th input token when generating the output.

*Mapping tokens back to code.* Token-level weights are difficult for humans to interpret directly. Therefore, we map tokens back to code units that match how developers naturally review diffs:

---

**Algorithm 1** Attention-based highlighting for diffs

1: **Input:** Prompt $P$ containing diff; chunking scheme $C$; budget $K$
2: Run LLM on $P$ to generate a one-token output and collect last-layer attentions
3: Extract attention tensor $A \in \mathbb{R}^{1 \times N + 1 \times H}$ for the generated token over input tokens
4: Pool over heads: $w_i \leftarrow \frac{1}{H} \sum_{h=1}^{H} A_{1,i,h}$ for $i \in \{1, \ldots, N + 1\}$
5: Map tokens to chunks (lines/hunks/files) using parser alignment (hierarchical token grouping)
6: Score each chunk as in Section 3.4.3
7: **Return:** Top-$K$ chunks by score

---

- **Line:** individual changed lines (fine-grained but may lack context).
- **Hunk:** contiguous blocks of change as shown by diff tools (contextual and commonly used in review).
- **File:** files touched by the diff (coarse-grained; useful for triage).

We compute a unit score by aggregating token weights within the unit and then return the top-$K$ scored units. In the UI, these are highlighted to direct developer attention. Section 3.4 goes into further detail on the grouping and aggregation approach.

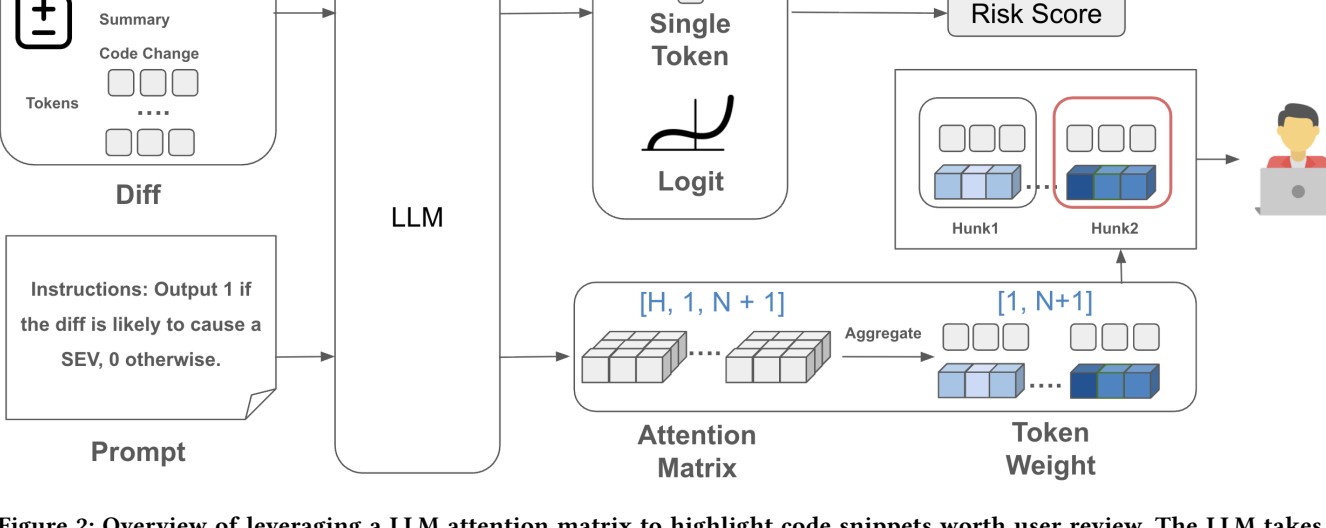

**Figure 2: Overview of leveraging a LLM attention matrix to highlight code snippets worth user review. The LLM takes a pre-defined prompt to instruct the model to distinguish the risk of a diff. The LLM generates a one-token sentence and the logits of the tokens — 0 and 1 — are used to infer the diff risk score. The attention matrix is aggregated to produce a $\{1, N + 1\}$ matrix representing each token's correlation towards the generated token. We further chunk the code into hunks and assign hunk importance by aggregating their tokens' attention weights.**

## 3.4 Hierarchical Token Grouping

LLM tokenizers decompose source code into subword units whose boundaries rarely coincide with the constructs that developers reason about during code review. A variable name such as `getUserById` may be split into three or more tokens, while a hunk header or filename may be scattered across dozens. To produce explanations that are actionable at the granularity of *tokens*, *lines*, *hunks*, and *files*, we introduce a four-level hierarchy that (i) reconstitutes subword tokens into semantically meaningful units, (ii) recovers the diff's structural boundaries from tokens, and (iii) recursively aggregates per-subword attention scores upward through the hierarchy with level-specific aggregation.

*3.4.1 Token Reconstitution.* The first stage maps a flat sequence of subword tokens and their associated scalar attention weights into *token groupings*—units that correspond to whitespace-delimited tokens in the original diff text. The DRS pipeline supports multiple backbone LLMs, so this logic is dependent on the tokenizer family.

*SentencePiece (word-boundary merging).* SentencePiece tokenizers mark word boundaries with a leading ␁ (U+2581) prefix. We scan the subword sequence left-to-right, accumulating consecutive subwords into a single token grouping until a new boundary prefix is encountered. The prefix character is stripped and a synthetic whitespace symbol is inserted between groupings so that downstream formatting can reconstruct the original text. The special byte-token `<0x0A>`, which SentencePiece uses for the newline character, is emitted as a dedicated newline symbol with a null score (not considered for attribution of risk).

*Tiktoken / BPE (pass-through).* Tiktoken-family tokenizers encode whitespace as part of the token text and do not use a word-boundary marker. Each raw token is therefore mapped one-to-one to a token grouping after splitting on literal newline characters. No multi-subword merging is required.

In both cases, the output is an ordered list of *token grouping* objects, each carrying the concatenated surface string and (the list of per-subword attention scores.

*3.4.2 Hierarchical Structure Recovery.* Given the flat sequence of token groupings, we reconstruct the diff's hierarchy in two passes.

*Line splitting.* Token groupings are partitioned into *line groupings* by detecting newline-delimiter tokens.

*Hunk boundary detection.* Within each file's code-change section, unified-diff hunk headers of the form @@ -a,b +c,d @@ are detected via regex. Each header initiates a new *hunk* object; subsequent lines are classified as additions (+), deletions (-), or context based on their leading character, and source/target line numbers are tracked incrementally from the header's start positions.

The result is a tree rooted at the diff, with interior nodes for files and hunks and leaves for individual lines and tokens (depicted in Figure 3).

*3.4.3 Multi-Level Score Aggregation.* Each node in the hierarchy carries a scalar attention score. Leaf scores are the raw per-subword weights produced by the LLM's attention heads; interior scores are computed by a recursive, bottom-up aggregation pass. Because the semantics of "importance" differ across levels—a subword's contribution to its parent token is qualitatively different from a hunk's contribution to its parent file—we employ *type-dispatched*

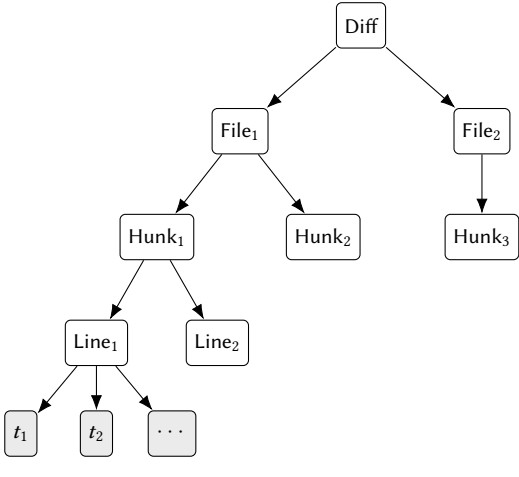

**Figure 3: Hierarchical grouping of diff attention scores. Subword tokens ($t_i$) are leaves; scores are aggregated bottom-up through lines, hunks, and files with level-specific functions.**

aggregation: for each node (e.g., *Hunk* or *Line*), a type-specific aggregation function is selected during the recursive traversal.

Formally, let $s(v)$ denote the score assigned to node $v$ and $\text{ch}(v)$ the ordered children of $v$. For a leaf token grouping with subword scores $(w_1, \ldots, w_m)$,

$$s(v) = \frac{1}{m} \sum_{i=1}^{m} w_i.$$

For an interior node at hierarchy level $\ell$ with aggregation function $f_\ell$,

$$s(v) = f_\ell\big(\{s(c) \mid c \in \text{ch}(v), \ s(c) \neq \texttt{null}\}\big).$$

Children whose scores are null (e.g., whitespace symbols or nullified test-file tokens) are excluded from the input multiset before $f_\ell$ is applied. If the resulting input is empty, $s(v) = 0$.

The top-$k$ sum used at the hunk level is defined as

$$f_{\text{hunk}}(\mathbf{x}) = \sum_{i=1}^{\min(k, |\mathbf{x}|)} x_{(i)},$$

where $x_{(1)} \geq x_{(2)} \geq \cdots$ is the sorted order of the children's scores.

#### 3.4.4 Practical Considerations.

*Escaped-newlines.* Certain tokenizers split the two-character escape sequence \\n across token boundaries (e.g., one token for \ and a separate token for n). A dedicated merge pass detects and rejoins these fragments before line splitting, preventing spurious line breaks that would corrupt hunk parsing. At the hunk level, line splitting accepts both literal and escaped newlines as delimiters to accommodate cross-tokenizer variation.

*Test-file score nullification.* Code changes in test files are often mechanically correlated with the code under test and rarely represent the *root cause* of a risky diff. We encode this domain prior by nullifying all token scores within files whose paths match any of a curated set of 16 regular expressions covering common test-file conventions across languages. Because null scores are excluded during aggregation, test files receive a score of zero at every level.

### 3.5 Implementation

Our implementation leverages the Hugging Face library[2], utilizing its unified API interface to ensure compatibility with other large language models (LLMs) [12]. The DRS service is implemented as a standalone Thrift API service[3], where clients can send diff information, including title, summary, test, and code changes.

*Explanation Calculation.* We embed diff information into a predefined prompt and instruct the LLM to identify outage risk and return a one-token response. By leveraging the logit of this token, we compute the risk score of the diff being risky (Section 3.3). We also obtain the attention tensor associated with generating that token and aggregate it to compute token-level weights. These weights are then used solely for highlighting; they do not affect the risk score.

*Tokenizer-agnostic pipeline.* The reconstitution and structure-recovery stages abstract over tokenizer-specific details behind a common *token grouping* interface. Adding support for a new tokenizer requires only a new conversion function that maps raw subwords to token groupings; downstream aggregation and explanation rendering are unchanged.

## 4 Experiment

### 4.1 Dataset

The method proposed in this work generates an explanation as a side product of risk score generation—it requires no additional model training. However, the DRS model is trained on data that does not include which lines caused outages. As such, standard DRS model evaluation data sets are insufficient to evaluate the quality of explanations. To conduct a quantitative evaluation, we created a an Outage Root Cause (O-RC) dataset that draws upon domain experts to pinpoint specific code changes that caused outages. The dataset was manually curated whereby each outage is tagged by four area experts. The main reviewer (typically the diff author) identifies outage-causing lines of the diff. The remaining three engineers perform validations. If the primary expert indicates high confidence, we include this diff in our dataset. All experts were instructed to find which lines of the diff were the most likely cause of the outage or leave the entry empty if it is not triggered by diff directly. The final dataset includes the file name and the code lines that caused the outage. For evaluation, we use our explanatory method to select the $K$ riskiest code snippets and then compute whether these snippets cover the root cause lines.

### 4.2 Evaluation Setup

**Coverage** This metric measures whether our method recommends results that contain the outage root-cause lines. Our model returns the $N$ code snippets with the highest score. We evaluate whether these snippets contain the actual outage-causing lines:

$$Coverage = \frac{\#\text{diff}_{\text{root\_cause} \in \text{explanation}}}{\#\text{diffs}}$$

---

[2]Hugging Face. (2022). Transformers: State-of-the-art natural language processing. Retrieved from https://huggingface.co/docs/transformers/index
[3]https://thrift.apache.org/

```
      @@ -76,10 +76,6 @@ async def get_version() -> AppVersion:
76    76
77    77    @app_router.get("/app_deps", operation_id="get_app_deps", status_code=200, response_model=AppDependencyVersions)
78    78    async def get_app_deps() -> AppDependencyVersions:
79      -     try:
80      -         xformers = version("xformers")
81      -     except PackageNotFoundError:
82      -         xformers = None
83    79    return AppDependencyVersions(
84    80        accelerate=version("accelerate"),
85    81        compel=version("compel"),
      @@ -93,7 +89,7 @@ async def get_app_deps() -> AppDependencyVersions:
93    89        torch=torch.version.__version__,
94    90        torchvision=version("torchvision"),
95    91        transformers=version("transformers"),
96      -         xformers=xformers,
      92  +         xformers=None,  # TODO: ask frontend
97    93    )
```

Figure 4: Illustration of hunk-level highlighting, which requests users to review 1 out of 2 hunks in a single file. The highlighted hunk contains 2 line changes.

**Average Review Percent (ARP)** Along with delivering high coverage, we seek to optimize developer time investment by asking them to review only the most important code snippets. The Average Review Percent within each diff is:

$$ARP = avg\left(\frac{\#\text{explanation\_lines}}{\#\text{diff\_lines}}\right)$$

We evaluate the proposed explanation method on the O-RC dataset to measure both attention-based explanation accuracy and associated developer workload. As discussed in Section 3.3, we chunk the diff into smaller snippets for code review. In this experiment, we explore which explanation granularity may work best in practice. We chunk the diffs into lines, hunks and files and then return the top 10/20/30 lines for line-level explanation, the top 1/2/3 hunks for hunk-level explanation, and the top 1/2/3 files for file-level explanation.

### 4.3 Results / Observations

The results of our study are presented in Table 1. The line-level explanation requires a minimal amount of code review. When we return the top-10 lines to developers, they only need to review 7.6% of the code to cover the outage-causing area with a coverage of 30.76%. As the line budget increases to 30 lines, developers have more opportunity to see outage-causing lines in the highlighted output, at the cost of additional scanning and context reconstruction.

When we return the top-1 hunk to the user, they must review 15.37% of the code (about 20 consecutive lines on average), which achieves 30.76% coverage. This is worse than the top-20 line-level explanation, which requires a similar review fraction. However, top-2 hunks achieve similar results to top-30 lines on both coverage and review amount. Since hunks tend to return larger blocks of code, an incorrect prediction of the top-1 hunk may lead to more time spent in review than an incorrect prediction at the line level.

For the file-level experiment, we only include diffs where not all the files touched by these diffs cause outages, *i.e.* not all the files are

part of the explanation and need to be investigated. File-level results demonstrate a trade-off: larger chunks yield higher coverage but incur substantially more review. When returning the top-1 file as the explanation, we have a coverage of 50% with a code review percent of 41.9% (55 lines on average). When we return the top-3 files, although we achieve 92% coverage, users must review the majority of each diff on average (78.61%). Based on these observations, we find that the top-2 hunks may be the most pragmatic granularity. Although line-level explanation has higher coverage when review cost is on-par with hunk-level, it can be too fine-grained to be easily comprehended in isolation. When highlighted lines are sporadically distributed throughout a diff, users must still read surrounding lines to understand context, implicitly increasing review effort. In contrast, file-level explanations are often too coarse grained for actionable debugging or validation. Hunk-level explanation provides a balance: hunks naturally segment diffs by context and often correspond to the unit of review in real-world workflows. By showing the top 2 hunks, we achieve 53.85% coverage with users reviewing 26.28% of each code change on average. We conclude that hunk 2 level provides the best balance of coverage, review lines, and review percentage.

Table 2 summarizes the aggregation choices used in production, which were selected empirically based on a held-out validation set. We rationalize the performance of top-$k$ aggregation as it captures concentrated signal (a few high-attention lines) while bounding the influence of very large hunks.

### 4.4 User Feedback

To collect feedback, we added our explanation method to the code review tool and provided a user interface to provide positive and negative feedback with free text entry. We received 55 pieces of feedback: 7 positive and 48 negative. Among the negative pieces of feedback, 17 contained textual info explaining why the user considered the explanation unhelpful. These comments consistently

**Table 1: Evaluation of Explanation on O-RC Dataset**

| Granularity | Threshold | Precision | Review Lines | Review Percent |
|---|---|---|---|---|
| **Line** | Top 10 | 30.76% | 10 | 7.60% |
| | Top 20 | 42.30% | 20 | 15.20% |
| | Top 30 | 53.85% | 30 | 22.8% |
| **Hunk** | Top 1 | 30.76% | 20.22 | 15.37% |
| | Top 2 | 53.85% | 34.57 | 26.28% |
| | Top 3 | 64.38% | 50.19 | 38.15% |
| **File** | Top 1 | 50.00% | 55.11 | 41.90% |
| | Top 2 | 69.23% | 79.68 | 60.50% |
| | Top 3 | 92.30% | 103.40 | 78.61% |

**Table 2: Aggregation functions by hierarchy level.**

| Level | Aggregation |
|---|---|
| Token | MEAN |
| Line | MEAN |
| Hunk | TOP-$k$ SUM (k=20) |
| File | MEAN |

point to one major shortfall of our current solution: highlighting risk for test cases and generated code. We also observed feedback indicating that highlights can be difficult to interpret when they emphasize boilerplate or common patterns rather than the specific change likely to trigger an issue.

As previous studies point out [17], though attention highlights the information source used to make a decision, the model can make decisions either toward or against the prediction outcome. This means that part of the highlighted code could represent important reasons why the diff is viewed as less risky. This feedback has inspired us to explore ways to identify attention signal polarity to further improve coverage.

## 5 Related work

The advent of exceedingly accurate AI models and their use in science and engineering has raised fundamental questions around what understanding means and how to attain it. Notably, AI models are accurate oracles, but do not, without additional technology, help humans understand their predictive rationale. One pragmatic solution from the philosophical standpoint [18] is that two conditions are satisfied:

(1) A phenomenon P can be understood if a theory T of P exists that is intelligible (and meets the usual logical, methodological, and empirical requirements).
(2) A scientific theory T is intelligible for scientists (in context C) if they can recognise qualitatively characteristic consequences of T without performing exact calculations.

In case of DRS predictions, the key question is to what extent the highlighted risky parts of the diff are "intelligible" to an engineer making decisions about the change.

According to Krenn *et al.* [19], an AI system can contribute to new scientific understanding as a "computational microscope" with the ability to acquire information not yet attainable through experimental means; as a "resource of inspiration" or an artificial muse, expanding the scope of human imagination and creativity; as an "agent of understanding", replacing the human in generalizing observations and transferring new scientific concepts to different phenomena, and conveying these insights to human scientists. In the case of DRS, the LLM without highlighting serves as a microscope showing which diffs may cause outages, but it becomes a more useful "agent of understanding" when it can also indicate where to inspect.

Explanation is key for understanding [20], and engineers and scientists are not satisfied without it. AI methods have a long history of dealing with explainability. From early rule-based expert systems through to decision trees, explainability typically took the form of revealing underlying rules and branch weights. Newer methods like random forest obscured the rules, but one could still estimate the importance of each feature used in prediction. Deep learning upended previous approaches to explainability. According to Xu *et al.* [21], the primary approach with deep learning models is post-hoc explanation: a result is inferred then an explanation is generated. Post-hoc explanation tries to (a) provide analytic statements; (b) show saliency maps via feature importance—in our case, to highlight a subset of a change; (c) give explanations by example. In the context of defect prediction, previous work includes fitting simple models to explain more complex models [22]. That might look like predicting risk with an LLM, but explaining those predictions with a logistic regression model.

A systematic review of the literature on explainable AI in software engineering [23] shows that defect prediction is the most common target, but that explainability is mostly investigated for traditional rather than deep learning methods [24]. Hence, our contribution is both relevant and timely. The survey finds that visualization and explanations in natural language are particularly uncommon in the field of explainable AI for software engineering.

Note, however, that the focus of this paper is to show that our approach is useful at scale at FAANGCompany, rather than a systematic comparison with other approaches.

## 6 Limitations

*Generalizability.* Generalizing from empirical studies in software engineering is difficult because of a large number of potentially relevant context variables. The study was performed at FAANG-Company, so the results might be different elsewhere. However, the software systems under study involve millions of lines of code and thousands of developers some of whom are collocated while others collaborate across multiple locations across the world. The systems also span a range of domains from social network products and virtual and augmented reality to software engineering infrastructure. While the desire to prevent outages is common among all (big tech) companies, the specific DRS tool we use may be unique to this study at present, but it could be of great value more broadly.

 

Our (preliminary) evaluation represents only a snapshot in time; the distribution may shift over time and for other domains or types of risk.

*Construct Validity.* We have used two measures: the existence of overlap with triggering entities and the percent of code highlighted. Even experts who annotated our diffs may not be 100% accurate as to what exact lines caused the outage; we plan to conduct future studies with more users to determine if the approach is effective in practice. We assume that highlighting helps developers identify and fix problems, but we have not yet measured downstream outcomes such as reduced time-to-resolution or reduced post-landing incident rates. We plan to conduct follow-up studies to understand how developers interpret highlighted snippets and how that interpretation affects decision-making.

*Internal Validity.* We make an assumption that the attention accurately reflects areas of the code that need inspection, but there may be better ways to indicate and explain the risk. We intentionally avoided some approaches we deemed too risky such as asking the model to provide natural language explanation that may produce hallucinations or factually incorrect claims. Additionally, attention weights may be influenced by prompt structure or formatting, and not solely by semantic properties of the change.

## 7 Conclusions & Future Work

This paper presented how LLMs are used at FAANGCompany to generate a score of how likely a diff will lead to an outage—a system we call Diff Risk Score (DRS) [1]. In this paper, by leveraging attention weights, we proposed an approach to highlight the parts of the change that most contribute to the predicted level of risk. We experimented at multiple levels: line, hunk, and file. To balance the coverage against the number of lines that an engineer needs to review, our initial results show that displaying the top-2 hunks to engineers is sufficient. We found a coverage of 53.85% while highlighting an average of 34 lines accounting for 26% of the total lines in a diff.

While our work is preliminary, our key contribution is the development and validation of an approach to explain just-in-time defect prediction via LLMs to professional engineers. This approach not only increases engineer trust in the model but also may help them mitigate risks associated with their changes. This has the potential to improve the reliability and trustworthiness of DRS models, enabling their broader deployment in a variety of industry settings. Furthermore, our attention-based explanations are highly scalable and efficient for real-world, large-scale software development workflows (such as the one at FAANGCompany).

Future work will focus on refining this approach, exploring its applications in other domains, and evaluating its impact on both software quality and developer productivity. Promising directions include incorporating additional signals (e.g., file history or test coverage) to reduce false highlights, and developing methods that distinguish attention supporting a high-risk decision from attention that supports the opposite.

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
