# OpenReview forum: "A Preliminary Study on Explaining Risk of Code Changes using LLM-based Prediction Models"
_ACM.org/AIWare/2026/Conference — AIware 2026_

### Official Review · Reviewer_GA4t · 2026-03-09

**Rating:** 1
**Confidence:** 4

**Review:**

The problem addressed by the paper is relevant and the proposed direction, using LLM attention mechanisms to guide developers toward risky parts of code changes, is interesting. However, the paper currently reads as an early draft and lacks sufficient methodological clarity and experimental rigor. Several key details required for reproducibility are missing, baseline comparisons are absent, and important parts of the system design are not sufficiently explained. Due to these issues, it is difficult to assess the validity of the claims or reproduce the results. Consequently, I recommend rejection in its current form.

**Strengths**

The paper addresses an important problem in software engineering: improving the interpretability of ML-based defect prediction models used in industrial development workflows. The idea of leveraging attention weights from LLMs to highlight risky code regions is interesting and potentially useful in practice. The industrial context and the manually curated outage dataset could also provide valuable insights if presented with sufficient methodological rigor.

**Weaknesses**
1. Although the idea and underlying technology are promising, the paper is poorly written and appears to be an early draft. Important explanations are incomplete, and several components of the proposed method are insufficiently described. This significantly harms readability and makes it difficult to understand the actual implementation details.

2. The paper omits several essential details required to reproduce the results. Key aspects of the experimental setup, including model configuration (temperature seed hyper-parameters etc.), inference settings (how many iterations?), and prompt design, are missing. Without this information, the work cannot be independently validated.

3. The evaluation does not include comparisons against any baseline methods. Given that the paper proposes an explanation technique, it should compare against existing explainability approaches such as attention-based methods, saliency methods, or feature-based explanation techniques used in reviewer-effort literature.

4. Section 3.2 (LLM-Based Diff Risk Score Model) requires substantial rewriting. The explanation of model selection, the training procedure, and the inference process is insufficient. The paper briefly states that a Llama-based model is used and produces a one-token response, but does not clearly explain the architecture choices, training configuration, or how logits are converted into risk scores in practice.

5. The paper explicitly states that the prompt used by the LLM is shown in Figure 1, but no prompt is actually provided in the figure or elsewhere in the paper. Since the model behavior heavily depends on prompt design, this omission makes it impossible to understand how the system operates.

6. The paper claims to derive risk scores directly from the logits of two output tokens. However, there is no evidence that the authors manually validated whether these logits correspond to meaningful risk predictions or whether they correlate with actual model confidence. Some form of sanity check or validation should have been performed.

7. The authors cite several papers ([13–16]) regarding attention-based explanations but do not explain how previous work used attention weights for interpretability or how the proposed method differs from those approaches. Simply listing citations is insufficient; a brief discussion is needed to situate the contribution within existing research.

8. Algorithm 1 appears in the paper without being explicitly referenced or explained in the surrounding text. Readers are not guided through the algorithm, and its relationship to the writeup flow remains unclear.

9. The paper claims that the proposed explanations are integrated into a code review UI, but the updated interface showing the highlighted snippets is not actually presented. The only figure shows a redacted version of an existing UI, which does not clearly demonstrate the proposed changes.

10. The paper introduces a manually curated dataset (O-RC dataset) but provides almost no descriptive statistics. Important details such as the number of diffs, number of outages, distribution of diff sizes, number of files per diff, and dataset splits are missing. Without such statistics, it is difficult to assess the scale of the study or the significance of the reported results.

11. Just-in-time defect prediction was never discussed after the abstract. The implications and usefulness of this approach should be more explicitly state, evaluated, and mentioned.

**Summary:**

This paper claims a preliminary study on explaining the risk of code changes in a just-in-time defect prediction setting using an LLM-based Diff Risk Score (DRS) model. The authors propose leveraging attention weights from a transformer model to highlight code regions (lines, hunks, or files) that potentially contributed to a predicted risk score. Attention scores are aggregated from token-level representations and mapped to higher-level code units to generate explanations for developers during code review. The approach is evaluated using an internally curated dataset of outage-causing diffs where domain experts annotated root-cause lines. The study measures coverage of root-cause lines and the proportion of code developers must review when following the generated highlights.

---

> ### Author Response · Authors · 2026-03-17
> **Rebuttal**
>
> Thanks for the detailed feedback. We respond to the main points.
>
> **Writing quality:**
> The strongest issues here are missing specifics (dataset stats, algorithm narration, etc.). We’ll address those directly in revision. We’ll also tighten wording where it currently sounds like an early draft.
>
> **Reproducibility / disclosure constraints:**
> Full reproduction is not possible due to proprietary models/data/infrastructure, as this study was done in the context of a FAANG. We will add an explicit limitations/reproducibility statement clarifying what can’t be disclosed and why (consistent with common industrial practice in applied SE venues).
>
> **Missing prompt in Figure 1:**
> The prompt is proprietary. We will make that explicit in the figure caption and explain what Figure 1 is intended to convey (pipeline/data flow), without implying the prompt is reproducible from the paper.
>
> **Logit validation:**
> The DRS risk score is a production signal validated against outage outcomes over time, not an ad hoc number. We will add a sentence and citations to prior DRS work to support this (e.g., the following ICSE paper https://ieeexplore.ieee.org/abstract/document/11121710).
>
> **Llama version ambiguity:**
> We will specify the exact model identifier/version: llama3-70B
>
> **Related work:**
> We will expand related work to clearly position against attention rollout, integrated gradients, learned explainers, and signed attribution methods, and highlight the deployment-driven trade-offs.
>
> **Algorithm 1 narration:**
> We will add a prose walkthrough that maps each step of the algorithm to the system pipeline.
>
> **Redacted UI figure:**
> We’ll improve the caption and textual description so readers understand what the highlighting looks like even with redaction.
>
> **O‑RC dataset stats:**
> We will add the missing dataset statistics and annotator agreement measures.
>
> **JIT defect prediction connection:**
> We’ll add a short bridge to the JIT literature (including recent critiques about edit semantics) and explain how diff-focused prompting/aggregation fits that context.

---

### Official Review · Reviewer_fpXb · 2026-03-11

**Rating:** 3
**Confidence:** 3

**Review:**

Personally, I think the main value of this paper lies more in its industrial relevance than in methodological novelty. I appreciate that the authors focus on a real challenge in production code review and present a lightweight explanation mechanism that could realistically be used in practice. At the same time, I remain somewhat cautious about the strength of the empirical claims, because the paper does not yet provide enough baseline comparisons or user evidence to fully justify stronger claims about trust, usability, or productivity gains.


- Quality: The problem setup is important, the method is coherent, and the evaluation is based in real incidents. However, the empirical evidence is still limited, especially for validating whether the proposed explanations are truly better than simpler alternatives.

- Clarity: Clarity is good overall. The paper is easy to follow, and the figures are helpful. A few inconsistencies and missing details reduce precision, but the main ideas are communicated well.

- Originality: Originality is moderate to low methodologically, since attention-based explanation is already a known direction. However, applying it to industrial diff risk prediction with hierarchical, diff-aware aggregation is still a useful practical contribution.

- Significance: The significance is moderate to high for practitioners and industry-oriented audiences. The paper addresses an important problem in production code review and provides insights that could be valuable beyond this specific deployment.


Pros
- The paper tackles an important real-world problem. A raw risk score is not actionable unless developers can see where to inspect.
- The proposed method is operationally attractive because it reuses attention already produced during inference, making the approach lightweight and deployment friendly.
- The hierarchical mapping from subword tokens to lines, hunks, and files is a sensible engineering contribution that matches developer workflows well.
- The evaluation uses expert-annotated outage root-cause data which strengthens the practical relevance.

Cons
- The main technical idea is somewhat incremental, and the paper does not sufficiently separate its contribution from prior work on attention-based saliency/explanation.
- There are no strong baselines, so it is difficult to judge whether the reported 53.85% coverage at 26.28% review effort is actually compelling.
- The dataset is not described in enough detail; the paper should report the exact dataset size, distribution, sampling process, time span, and inter-annotator agreement.
- The user-feedback results are concerning, with most feedback being negative, which weakens claims about improved trust and usability.
- The current metrics are limited; binary overlap-based coverage does not fully capture ranking quality, partial recall, or explanation faithfulness.
- Some claims are slightly stronger than the evidence supports, particularly around trust and productivity benefits.

**Summary:**

Overall, this is a practically relevant and clearly written paper. The problem setting is important: a risk score alone is not very actionable unless developers can also see where to inspect in the code change. The method is coherent, easy to understand, and well aligned with real deployment constraints because it reuses attention already produced during inference. The evaluation on real outage-related incidents is also a meaningful strength.

---

> ### Author Response · Authors · 2026-03-17
> **Rebuttal**
>
> Thanks for the assessment. Responses to the main concerns:
>
> **Novelty / separation from prior work:**
> The novelty here is not “attention is useful” in the abstract; it’s the diff-aware aggregation pipeline that maps subword attention to line/hunk/file units for a production JIT defect prediction workflow with expert-annotated outage data. We will expand related work to position this against rollout/gradient methods, learned explainers (e.g., ExpNet), and signed attributions (e.g., CA‑LIG), and explain why we used inference-time aggregation (deployment constraints: latency + gradient access).
>
> **Baselines:**
> We will add dataset-derived reference baselines (random-at-effort, largest-hunk-first) so the 53.85% / 26.28% point can be interpreted without extra infrastructure changes.
>
> **Dataset description:**
> We will add: number of diffs, language mix, time span, diff-size/file-count distributions, annotator process, and agreement (Cohen’s κ / Krippendorff’s α), plus confidence intervals.
>
> **User feedback:**
> We will rewrite this section to be precise: early qualitative feedback (n=55, 7 positive), tied to specific failure modes. We’ll treat it as debugging signal, not as a user study.
>
> **Metric limitations:**
> We will add discussion of richer alternatives (Recall@k at line level, coverage–effort curves, AUCU) and why we used the current metric (it corresponds to a binary “inspect this hunk first” decision).
>
> **Overstated claims:**
> We will revise abstract/intro/conclusion to remove any suggestion that trust/productivity benefits were demonstrated, and instead treat them as motivation / future evaluation.

---

### Official Review · Reviewer_KmS7 · 2026-03-11

**Rating:** 3
**Confidence:** 4

**Review:**

## Strength
+ They bring attention-based localisation into just-in-time defect prediction settings, and move beyond diff-level risk scoring toward code-region guidance.
+ The problem is relevant and practical. If successful, this direction could make risk prediction much more actionable during code review.
+ The manuscript is generally easy to follow, and the pipeline from token attention to line, hunk, and file highlighting is explained clearly.

## Weakness
- The actual contribution is narrower than its framing. It more convincingly demonstrates an attention-based localisation heuristic than a validated interpretability advance.
- The paper claims improvements in trust, usability, and actionability, but does not measure those outcomes directly, and the only user feedback reported is mostly negative.
- The empirical case is weakened by the absence of baselines, an underspecified and likely small evaluation set, a coarse overlap metric, and strong built-in assumptions such as test-file nullification.
- Key details needed to evaluate robustness or reproduce the study are missing, including the exact O-RC sample size, the held-out validation procedure, and enough implementation detail for independent replication.



## Detailed Comments

### Novelty
I found the paper’s main idea interesting and practically motivated. In particular, the attempt to turn model attention into a lightweight review aid for large diffs is a reasonable and useful direction, especially in an industrial setting where low-latency solutions matter. The contribution is also more concrete than many high-level interpretability discussions.
That said, I think the current framing slightly overstates what has been shown. The paper presents the approach as an explanation method, but the evidence more directly supports a narrower claim: that attention can be aggregated into review-oriented units and used to prioritise where engineers might inspect first. That is still valuable, but it is different from demonstrating faithful or human-validated interpretability.
### Significance

The practical importance of the problem is clear. A diff-level risk score is often hard to act on without some guidance about where the concern may lie, particularly for large or complex changes. In that sense, the paper addresses a meaningful gap between prediction and actionability.

My hesitation is mainly about how strongly the practical impact is stated. The paper suggests that the approach may improve trust, usability, and actionability, but the actual evaluation focuses on overlap with expert-labelled root-cause lines and the fraction of code that must be reviewed. Those are useful proxy measures, but they do not directly demonstrate that engineers trust the system more, review more efficiently, or make better decisions because of it.

The user feedback section is especially important here. The fact that the deployment feedback was largely negative does not negate the value of the work, but it does suggest that the present system may still be at an early stage from a user experience perspective. I would therefore encourage the authors to frame the contribution as evidence of promise rather than as evidence of established practical benefit.

## Soundness
This is the area where I think the paper would benefit most from strengthening.

The biggest gap, in my view, is the lack of baseline comparisons. The reported results are interesting, but without comparison to simpler alternatives, it is difficult to judge how much of the gain comes from the proposed attention-based method itself. For example, it would be useful to know how the approach compares against simple heuristics such as prioritising larger hunks, non-test hunks, or historically risky files, or against another attribution method.

A second concern is that the evaluation set is not described with quite enough precision. Section 4.1 explains the construction of the O-RC dataset, but the exact final evaluation size is not clearly stated in the main discussion. Given that the paper reports fairly specific percentage differences, it would help readers interpret the results if the authors reported the exact number of evaluated diffs, along with some indication of uncertainty.

I also think the current coverage metric has limitations that should be discussed more directly. At the hunk level, a prediction is counted as successful if the highlighted hunk contains the expert-marked root-cause line. This is reasonable for a first analysis, but it can reward broad overlap rather than precise localisation. As a result, the numbers may look stronger than the actual localisation precision felt by a reviewer.

Another point is that the system incorporates a strong modelling assumption by nullifying token scores in files identified as tests. This may be entirely reasonable for the intended deployment setting, but it means the method is not purely exposing model attention. It is partly a post-processed localisation pipeline shaped by domain assumptions. I would encourage the authors to discuss this more explicitly, and perhaps frame it as a pragmatic design choice rather than a neutral explanatory mechanism.

Finally, the paper does a good job of acknowledging that attention may be directionally ambiguous, meaning that highlighted code may not necessarily support the high-risk prediction. I appreciated that honesty. At the same time, this issue has important implications for how the highlighting should be interpreted by engineers, and I think the paper could engage with that consequence more directly.

### Presentation

Overall, I found the paper readable and logically structured. The pipeline is well broken down, and the progression from motivation to model signal extraction to evaluation is easy to follow. The figures are also helpful in making the method concrete.

My main suggestion here is to soften some of the claims so that they align more closely with the evidence presented. In a few places, particularly in the introduction and conclusion, the paper implies stronger human-facing benefits than the current evaluation can support. I do not think this requires major rewriting, just slightly more careful phrasing.

There are also a few terminology and wording issues that should be cleaned up. The most important one is the mismatch between the metric name in the text and the label used in Table 1. There are also a few awkward phrases and template remnants that make the draft feel less polished than the core work deserves.

### Transparency/Reproducibility

I appreciated that the paper is fairly transparent about its limitations. In particular, the discussion of attention as a proxy rather than a causal explanation is thoughtful and important.

At the same time, several details needed for reproducibility or careful assessment are still missing. The exact evaluation sample size should be reported clearly. The held-out validation procedure used to select aggregation choices should be described in more detail. It would also help to provide more information about the test-file filtering rules and the specific implementation choices in the aggregation pipeline.

I understand that full artefact release may not be possible in an industrial context. Even so, the paper could still do more to support partial reproducibility through pseudocode, fuller parameter descriptions, clearer dataset counts, and additional ablations.

## Minor Comments
1. Section 4.1: “we created a an Outage Root Cause (O-RC) dataset” -> “we created an Outage Root Cause (O-RC) dataset.”
2. Section 4.2 vs. Table 1: the metric is defined as Coverage, but Table 1 labels the corresponding column Precision.
3. Section 4.3: “We conclude that hunk 2 level provides the best balance...” - > “the top-2 hunk setting” or “the hunk-level setting with top-2 hunks.”
4. Figure 2 caption: “The LLM generates a one-token sentence...” ->“One-token output” or “one-token response”.
5. Section 4.3: “too coarse grained” -> “too coarse-grained.”

**Summary:**

This paper presents a preliminary industrial case study on making LLM-based just-in-time defect prediction more interpretable during code review. The authors build on a Diff Risk Score system that predicts whether a code change may cause an outage, and instead of only showing a risk score, they use the model’s attention weights to highlight the parts of the diff, at the line, hunk, or file level, that the model focused on most when assigning risk. Their goal is not to claim a causal explanation, but to give developers a lightweight, low-latency pointer to where they should inspect first in a large change. Using an expert-annotated dataset of real outage-causing diffs, they find that highlighting the top-2 hunks gives the best trade-off, covering 53.85% of expert-marked root-cause lines while asking reviewers to inspect only 26.28% of changed lines on average. The study argues that this makes LLM-based risk prediction more actionable and scalable in practice, while also acknowledging key limitations, especially that attention is only a proxy for explanation and that users found the method less helpful when it highlighted test code, generated code, or signals whose polarity was unclear.

---

> ### Author Response · Authors · 2026-03-17
> **Rebuttal**
>
> Thank you for the careful review. We respond point-by-point.
>
> **Framing (heuristic vs interpretability method):**
> We don’t think “heuristic” vs “interpretability method” is a clean or particularly useful distinction in applied settings. What we claim is narrower: aggregated attention, mapped to code hunks and evaluated against expert-annotated outage root causes, provides a practical highlighting signal that is measurably better than the current production baseline (no highlighting). We do not claim causal faithfulness, and we will tighten the intro/conclusion language accordingly (i.e., “pragmatic highlighting validated against expert root-cause annotations”).
>
> **Attention faithfulness:**
> Yes: last-layer, head-averaged attention to one token is an imperfect proxy and can be prompt-sensitive. For our use case, the question is whether it helps developers triage—not whether it is a faithful causal attribution. The reported numbers (53.85% coverage at 26.28% review burden) are the empirical evidence we rely on. We’ll add a short discussion contrasting this with more “faithful” methods (Integrated Gradients, rollout, signed attribution like CA‑LIG) and why we didn’t use them here: latency constraints and lack of gradient access in parts of the deployment stack.
>
> **Baselines:**
> The baseline in production today is a scalar risk score with no highlighting, and our method is strictly more actionable than that. We agree that simpler controlled heuristics would help contextualize results; we’ll add reference baselines we can compute from the existing dataset (e.g., random line/hunk selection at 26% effort; largest-hunk-first) and discuss what they imply.
>
> **Evaluation set size / uncertainty:**
> We will add the O‑RC dataset size prominently and include inter-annotator agreement plus confidence intervals for the reported coverage.
>
> **Coverage metric:**
> We intentionally measure overlap at the hunk level because that matches how diffs are reviewed in practice (at least in our industrial context). This does make the metric coarser; we’ll make the precision/effort trade-off explicit and clearly state what this metric does and does not capture.
>
> **Test-file nullification:**
> This should be described as a pragmatic engineering choice, not as “pure attention.” We’ll revise the text to separate (a) the raw attention-derived signal from (b) post-processing rules, and we’ll be explicit that the remaining test/codegen failure cases suggest our current regex-based filtering is incomplete and should incorporate stronger signals (build metadata, codegen signatures).
>
> **User feedback:**
> We will tone down any language that reads like a broad usability validation. The feedback we report is early-stage qualitative input that surfaced concrete failure modes (tests, generated code, directional ambiguity). We’ll frame it as “known limitations under active work,” not as a claim of overall productivity impact.
>
> **Minor issues:**
> We will fix the identified table mismatch and phrasing/grammar issues.

---

> > ### Comment · Reviewer_KmS7 · 2026-03-20
> >
> > Thank you for the thoughtful and constructive rebuttal. I appreciate the clear responses and the willingness to narrow the framing. The clarifications around positioning the contribution as a pragmatic highlighting signal, softening stronger trust/usability claims, reporting the O-RC dataset size more clearly, adding confidence intervals, and separating raw attention from post-processing choices are all helpful and address several of my concerns.
> >
> > The main issue that still feels only partially resolved for me is the baseline story. I appreciate the plan to add simple reference baselines, and I agree that comparing against the current production setup with no highlighting is useful. At the same time, my original concern was not only whether highlighting is better than none, but whether the proposed attention-based method is well justified relative to simpler alternatives. So I still think the paper should be careful in how strongly it interprets the gains unless those additional baselines show a clearer advantage for the proposed approach.
> >
> > I also appreciate the clarification around the hunk-level coverage metric and the plan to state more clearly what it does and does not capture. That helps. Relatedly, I think the paper will be strongest if it keeps the overall framing modest and presents the results as evidence of promising practical localisation utility, rather than broader validation of interpretability or usability.
> >
> > My initial assessment was already on the positive side, so I am not changing the rating, though the rebuttal does improve my confidence in the paper’s framing and planned revisions.